# Harnessing the Wikipedia Graph for Effective Multi-Entity Question Answering

## Abstract

Wikipedia serves as a rich repository of well-curated knowledge, making it a popular source for information retrieval through question answering (QA). Often, these inquiries involve multiple entities, such as "How many Turing Award winners are Canadian?", necessitating the consolidation of information from various Wikipedia pages. Multi-entity question answering typically comprises two steps: multi-entity retrieval and subsequent reasoning using large language models (LLMs). The pre-defined connections within Wikipedia, known as the wiki-graph, facilitate relatively straightforward multi-entity retrieval. However, traditional solutions leveraging retrieval-augmented generation (RAG) encounter limitations, as LLMs often struggle to aggregate insights from multiple pages effectively. In response, we propose a Structured QA (SQA) approach that first organizes extracted entities into a relational table (e.g., a table schema with columns `(name, nationality)` for Turing Award winners) and then employs table-based methods such as TableQA or NL2SQL for answering. Extensive experiments demonstrate the superior effectiveness of SQA in addressing multi-entity QA challenges over Wikipedia, improves the overall accuracy 29.6% over the SOTA solutions, paving the way for more robust information retrieval from Wikipedia.

## 1 Introduction

Wikipedia (Wikipedia web, 2004) serves as a crucial and meticulously curated repository of global knowledge, offering users a comprehensive understanding of diverse topics. A common method for extracting information from this extensive resource is through question answering (QA) (Chen et al., 2017), often exemplified by inquiries like, "What are the capitals of countries that border France?" or "How many Turing Award winners are Canadian?" These questions frequently involve multiple entities, highlighting the intricate nature of information retrieval in today's data-rich environment.

The significance of multi-entity QA lies in its ability to sift through vast datasets to uncover hidden insights. Addressing questions that span multiple entities, not only expands the scope of retrievable information but also deepens users' understanding of complex relationships between entities. Furthermore, as the volume of available data skyrockets, multi-entity QA becomes increasingly vital for enhancing user interactions with extensive knowledge bases, ultimately transforming the way individuals access and utilize information. The primary challenges in multi-entity question answering (QA) involve the accurate parsing and understanding of complex queries to deliver precise responses. This necessitates the integration of advanced Natural Language Processing (NLP) techniques to effectively address these issues. With the ongoing development of large language models (LLMs), LLM-based approaches have recently emerged, primarily focusing on question-answering tasks using vanilla LLMs or incorporating Retrieval-Augmented Generation (RAG) (Gao et al., 2023).

**Existing Approaches.** A straightforward approach involves directly prompting vanilla LLMs with the given questions to obtain responses, as illustrated in Figure 1-b1. However, LLMs frequently encounter difficulties in delivering accurate answers due to several factors, including hallucination (Ji et al., 2023), outdated information, etc. A prevalent one-step further method is to incorporate RAG, retrieving relevant Wikipedia pages prior to inputting the extracted content along with the query

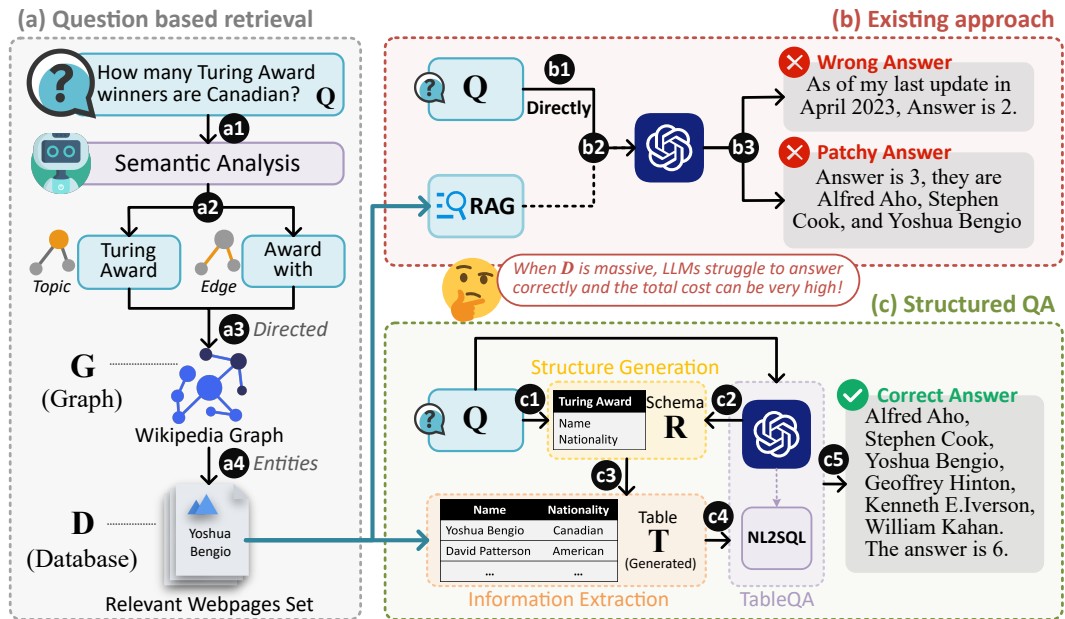

Figure 1: An Overview of Multi-entity QA Solutions over Wikipedia Graph. (a) Multi-entity Retrieval. (b) Existing Solution. (c) Our proposal: Structured QA.

into the LLM, as demonstrated in Figure 1-b2. Nevertheless, LLMs continue to face challenges in effectively synthesizing insights from multiple pages, resulting in incorrect or patchy answers.

**Our Proposed Structured QA.** To tackle the challenge of LLMs in aggregating information from multiple sources, we propose the Structured Question Answering (SQA) approach, as illustrated in Figure 1(c), to effectively address multi-entity QA. Specifically, we first infer a relational schema $R$ based on the user query $Q$. For example, we derive the schema (name, nationality) from the query "How many Turing Award winners are Canadian?" Next, we perform a graph search over a pre-defined Wikipedia graph to retrieve relevant entities. The information from these entities is then processed, extracted, and organized into a structured table $T$. With this table in hand, we can input it along with the query into an LLM, which is proven significantly enhances the ability in multi-entity QA tasks.

**Contributions.** Our notable contributions are summarized as follows.

- **MEGraphQA: Specialized Benchmark for Multi-Entity QA.** We curate MEGraphQA, a standardized framework based on Wikipedia for evaluating the effectiveness of various approaches in addressing the complexities of information extraction and reasoning involving multiple entities.

- **SQA: Innovative Approach to Structure Entity Information.** We propose the SQA approach, a novel method for managing vast and unstructured data by extracting properties of entities and organizing this information into structured tables. This method transforms textual information about entities into a format that facilitates analysis. Our experiments demonstrate its remarkable performance, achieving SOTA results and outperforming the strongest baselines by 29.6% in overall accuracy, while leading across all eight subtasks.

- **Enabling Sophisticated Data Analysis Techniques in Wikipedia QA.** Our proposed SQA explores the integration of advanced data analysis techniques in question answering (QA). The adoption of a structured data format facilitates the implementation of more sophisticated algorithms, enabling precise identification of pertinent information that may be difficult to extract from unstructured text. This fosters future research by offering a new perspective.

## 2 RELATED WORK

### 2.1 RETRIEVAL-AUGMENTED GENERATION

Advanced Retrieval-Augmented Generation (RAG) systems have evolved to incorporate pre-retrieval, retrieval, and post-retrieval strategies aimed at mitigating the limitations of basic RAG methods. In parallel, Modular RAG systems have introduced frameworks for iterative and dynamic cycles of intertwined retrieval and generation processes (Gao et al., 2023). Community summaries serve as a form of self-memory (Cheng et al., 2023), enhancing generation-augmented retrieval to support future cycles of information generation. Additionally, the parallel generation of community answers from these summaries reflects an iterative (Shao et al., 2023) or federated (Wang et al., 2024b) retrieval-generation approach. Similar concepts have been amalgamated in other systems for tasks such as multi-document summarization (Su et al., 2020) and multi-hop question answering (Feng et al., 2023). Furthermore, using a hierarchical index and summarization aligns with alternative methods, such as generating a hierarchical index of text chunks through vector clustering (Sarthi et al., 2024) or constructing a "tree of clarifications" to address multiple interpretations of ambiguous questions (Kim et al., 2023).

### 2.2 GRAPH QA

A large body of research has emerged at the intersection of graph-based techniques and LLMs (Pan et al., 2024). This exploration spans different aspects, from the design of generic graph models (Edge et al., 2024) and multi-modal architectures (Yoon et al., 2023) to practical applications. Notable applications include basic graph inference (Chai et al., 2023), node classification, graph classification/regression (Chen et al., 2024), and leveraging LLMs for knowledge graph-related tasks (Jiang et al., 2023). The domain of Graph Question Answering (Graph QA) is witnessing a notable shift with the introduction of emerging methodologies that apply Probabilistic Entity-centric Fine-tuning (PEFT) to graph-based Large Language Models (LLMs). This innovative approach is exemplified in the creation of models such as GraphLLM (Chai et al., 2023) and GraphToken (Perozzi et al., 2024), which are specifically designed to tackle basic graph reasoning tasks. Additionally, the development of GNP (Graph Neural Prompting) (Tian et al., 2023) marks a pivotal enhancement for multi-option QA on knowledge graphs.

### 2.3 BENCHMARKS FOR GRAPHQA

Comprehensive benchmarks specifically tailored for graph modalities lack extensive research. In contrast to existing benchmarks that focus on fundamental graph-based reasoning tasks such as node degree, edge existence, and shortest paths (Wang et al., 2024a) (Fatemi et al., 2023), G-retriever's benchmark (He et al., 2024) is applicable to complex and real-world graph applications, including commonsense reasoning, scene understanding, and knowledge graph reasoning. This fills a critical gap in evaluating the progress of models, but these models are not sufficient to solve various problems in graph-based applications. Based on this, we propose comprehensive Multi-Entities Graph QA benchmark (MEGraphQA) and provide a first-of-its-kind original model (SQA).

## 3 PROBLEM STATEMENT

### 3.1 WIKIPEDIA GRAPHS

The graph is developed by extracting the structural framework of Wikipedia, and Wikidata serves as the lexicon for the assignment of unique identifiers to each entity.

**Wikipedia Graph.** A wikipedia graph is represented as $G = (V, E, P)$, where $V = \{v_1, v_2, ..., v_n\}$ is the set of nodes in the graph, with each node $(v_i \in V)$ representing an entity; $E$ is the set of direct edges in the graph, with each edge $(e_j(v_i, v_k) \in E)$ representing a connection (or relationships) between two nodes. An edge is a tuple $(v_a, v_b, t)$, where $(v_a, v_b \in V)$ and $t$ is the type of the relationship. For example, $E = \{(v_1, v_2, t_1), (v_2, v_3, t_2), ..., (v_m, v_n, t_k)\}$. $P$ represents the set of properties associated with both nodes and edges. For nodes and edges, we can have $P(v_i)$ and $P(e_j)$ as the sets of properties associated with nodes $v_i$ and edge $e_j$, respectively. Each property can be

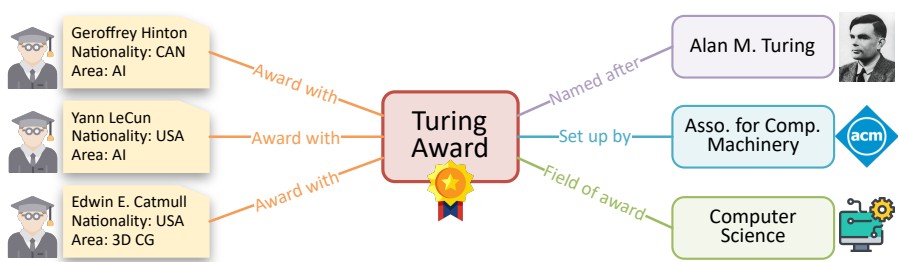

Figure 2: Illustration of Wikipedia graph fragment.

represented as a key-value pair $(k, va)$, where $k$ is the property key, and $va$ is the property value. Figure 2 is a sample of a Wikipedia graph snippet.

**Hops.** In terms of Wikipedia Graph systems, the term 'hop' refers to a step taken along the edges of a graph from one node to another, so we will consider 'hop' as a tuple $(v_a, v_b, t)$. For no hop, there is no relationship (edge) to track. Single hop track $(v_a, v_b, t)$ and find all entities has relationship $t$ to $v_a$. Multi-hop involves traversing multiple edges (hops) to find connections between nodes that are not directly linked, like track $(v_a, v_b, t_1)$ $(v_b, v_c, t_2)$.

### 3.2 Queries over Wiki Graph

**Multi-Entity Query.** Multi-entity queries are common and necessary in many application scenarios. For instance, "How many ACM Fellows work on machine learning?". This query requires aggregation of information across the entities related to compose an answer. Let's define multi-entity queries $Q_m = L(Y, V, P_v)$, $L$ is the LLM for query generation. $Y$ is the types of queries, $Y = \{y_1, y_2, ..., y_n\}$. $V$ is the entity set in which the user is interested.

**Query Categorization.** To design effective query types for a Property Graph QA system, it is important to consider the nature of the data stored in the property graph and the types of queries that users are likely to ask. The types should cover various query intentions, We categorize queries into three types: (1) comparison (2) statistical (3) relationship exploration. The detailed explanation will be expounded in Section 5.

## 4 SQA System Design

In our SQA system, we integrate several fundamental components, including graph search, table generation, and an executor, to provide users with a comprehensive platform for the exploration and analysis of graphs. It enables the transformation of entity properties to address specific queries and facilitates advanced semantic and relational operations, thereby allowing users to extract valuable insights from the data effectively and efficiently.

The initial step involves conducting a **graph search** across the Wikipedia graph to identify relevant entities. Secondly, **table generation** begins with "guessing" a table schema based on the given query, followed by the extraction of information from the identified entities to populate the table. Finally, we implement an **executor** that processes the generated table to respond to the specified query. Next, we will elaborate on the details of each step.

### 4.1 Graph Search

Graph search in Wikipedia involves employing graph theory and algorithms to navigate the interconnected network of articles and links. We deploy GPT-4 as the Semantic Analysis Model to identify entities and properties within user queries, as illustrated in Figure 1-a2. Subsequently, we implement Named Entity Recognition (NER) to detect and classify named entities referenced in the queries, alongside applying Relation Extraction (RE) techniques to extract relevant properties that serve as edges for each identified entity. For multi-hop queries, the Semantic Analysis process

initially deconstructs the queries into sub-queries, allowing NER and RE to be applied sequentially to each sub-query to identify named entities and extract properties until all sub-queries have been processed.

## 4.2 TABLE GENERATION

The table generation consists of two steps: (1) generate the schema and (2) extract entity to fill the table.

**Schema Generation.** We use GPT-4 to generate properties for the graph nodes which can be interpreted as the table schema to construct the table, facilitating responses to queries. These generated properties may necessitate adjustments or refinements to align more closely with the intent of the input query. For example, for the query "How many Nobel Prizes in Physics laureates have been awarded for discoveries in Particle Physics?", the LLM produces a schema (name, YearAwarded, field), there are two issues, first is that (field) is oversimplified, it should be (field in Physics). the second issue is the column YearAwarded are redundant. Therefore, we indicatively prompt GPT-4 to review critically to reduce three issues as listed below, and the prompt is in Appendix A.1.1.

- Over-Simplification: Ensure the properties are not too simplistic for the complexity of the question it needs to support.
- Missing Elements: Make sure no relevant component is overlooked. Every aspect of the question should be answerable with the properties.
- Redundancy: Avoid unnecessary properties (columns) or information that could complicate data analysis.

**Entity Extraction.** In our data processing workflow, we extract relevant information from the graph RAG data utilizing more cost-efficient LLMs like Mistral-7B. The extracted information is then used to populate a generated table, where each row corresponds to a unique entity from the graph RAG data, as illustrated in Figure 1-c3. Ultimately, we transform information in entities into structured tables, facilitating subsequent access, manipulation, and interpretation for various analytical purposes.

## 4.3 EXECUTOR

Queries are received in natural language. Because of the ambiguity of natural language (Liu et al., 2024b), the input query needs to be translated into a structured query language (e.g., SQL) that the system can execute on the generated table. While LLM (like GPT-4) generates the table schema, SQL is also generated. We leverage the SQLite3 library (SQLite Development Team, 2023) to manage and interact with SQLite databases.

## 4.4 OPTIMIZATION

Three aspects of optimization are included in SQA system to enhance the overall performance:

**Data Filtering**. One significant advantage of the process of extracting information from text and organizing it into a tabular format is the inherent ability of this process to filter out irrelevant data. During the extraction phase, LLMs are employed to identify and categorize the most pertinent pieces of information based on schema. This means that only data deemed relevant to the query is selected for inclusion in the table. As a result, users are presented with a clean, concise table that contains only the data necessary for their analysis or decision-making processes.

**Model Selection.** Model selection is straightforward yet highly effective for optimization (Liu et al., 2024a). The SQA system comprises multiple tasks, necessitating the selection of the most suitable model for different tasks. For basic tasks, more affordable and faster LLMs can suffice, while utilization of the most advanced LLMs is essential in more complex tasks to ensure optimal performance. Specifically, SQA system employs powerful yet resource-intensive GPT-4 for tasks such as semantic

analysis or generation of table schemas and SQL queries. In contrast, for more basic information extraction, we utilize open-source Mistral-7B, thereby achieving a balance between cost efficiency and functional performance.

**LLM Input/Output Control**. SplitWise (Patel et al., 2023) shows that LLM inference time is generally proportional to the size of input and output tokens. Since GPT models decide the cost based on the input token, we try to minimize the input of large models. Meanwhile, we use the instructive prompt to reduce the size of the outputs generated by LLM without changing the quality of these outputs. The example of prompt is in Appendix A.1.2.

## 5    MEGRAPHQA BENCHMARK

### 5.1    DESIGN OBJECTIVES AND SCOPE

Multi-entity questions extend the focus to consider the interactions and relationships between multiple entities simultaneously. We introduce MEGraphQA, covering three types of questions:

- **Comparison questions** are primarily concerned with examining differences or similarities between groups, entities, or time points, which involve comparing the performance of two different groups under varying conditions or assessing changes in a particular variable over time.
- **Statistical questions** (Zhu et al., 2024) delve deeper into the data, seeking to understand the distribution, trends, and patterns that emerge within a dataset. These questions often involve the application of statistical methods to infer properties about the population from which the data was sampled, such as mean, variance, and correlation coefficients.
- **Relationship questions** aim to uncover the connections and associations between different variables within the dataset, including exploring how one variable may influence another, identifying causal relationships, or mapping out complex interactions among multiple variables.

This categorization facilitates a more targeted approach to problem-solving, acknowledging that applicable methodologies may vary considerably. Table 1 illustrates examples of multi-entity queries, organized into three categories with eight further sub-types. A total of 4,780 questions have been allocated across two datasets: the test set and the training set. The test set comprises approximately 30% of the total questions, amounting to around 1,374, while the training set contains the remaining 3,406 questions. Table 2 provides statistics of the MEGraphQA benchmark.

### 5.2    AUTOMATED QA GENERATION AND VALIDATION

We extract the introductory paragraph of textual content for each entity from Wikipedia, akin to an abstract of the entity's page, to derive relevant property values. The preprocessing of graph node properties is conducted using GPT-4. GPT-4 is deployed to generate the essential properties of key entities for each topic, and subsequently, property values are extracted from the respective web pages of these entities. This process culminates in the formation of property tables. An illustrative example of the topics and entities' properties is provided in Appendix Table 4.

When questions or queries are posed, the SQA system efficiently navigates the graph by utilizing both the connections (edges) and the nodes along with the associated property tables to retrieve relevant information. The property tables, which contain attributes and values related to the entities within the graph, serve as a comprehensive and structured data source that can be queried alongside the graph structure. This dual approach facilitates thorough analysis, as it takes into account both the relational context (the connections among entities) and the specific properties of the entities involved. Moreover, such automated process benefits from low labor costs due to automation and optimization within the graph database system, reducing the need for time-consuming and error-prone manual data processing and analysis.

### 5.3    QUALITY CONTROL

We devise several strategies to ensure the integrity and effectiveness of questions.

Table 1: Examples of multi-entities queries.

| Types | Sub-types | Examples |
|---|---|---|
| Comparison | Intercomparison | Which country has more ACM fellow, China or USA? |
| | Superlative | Which city has the highest population? |
| Statistics | Aggregation | How many ACM fellow are from MIT? |
| | Distribution Compliance | Does the nationality of ACM fellows follow a normal distribution? |
| | Correlation Analysis | Is there a linear relationship between total number of events and records broken in all Olympic Games? |
| | Variance Analysis | Are the variances in number of participating countries and total number of events of Summer Olympics different significantly? |
| Relationship | Descriptive Relationship | If there is a relationship between the year of induction as an ACM fellow and the fellows' areas of expertise? |
| | Hypothetical Scenarios | Will China overtake the US in the gold medal tally at the 2024 Paris Olympics if it wins one more gold medal? |

Table 2: Statistics of MEGraphQA benchmark.

| Categories | MEGraphQA-train | MEGraphQA-test | MEGraphQA-total |
|---|---|---|---|
| #-Queries | 3406 | 1374 | 4780 |
| #-one-hop Q | 1406 | 606 | 2012 |
| #-multi-hop Q | 1322 | 768 | 2090 |
| Ave. #-entities /Q | 460 | 391 | 409 |
| #-Topics | 165 | 76 | 241 |
| #-Comparison | 1107 | 438 | 1545 |
| #-Statistics | 1440 | 585 | 2025 |
| #-Relationship | 859 | 351 | 1210 |

- **Question Templates.** The use of templates ensures that every question is crafted with a clear structure, making it easier for respondents to understand and answer them accurately. For relationship and complex statistic questions, we turn the questions in a closed-ended style, as they require a specific response of either "yes" or "no", which makes the answer in a standardized format. We meticulously prepare all question templates, with examples in the Appendix Table 5.

- **Question Refinement.** After the initial development phase, each question undergoes a refinement process utilizing GPT-3.5-turbo. This stage is essential for improving the clarity, relevance, and neutrality of the questions. It also includes a thorough review to identify and mitigate any potential bias, contributing to minimizing misunderstandings and elevating the overall quality of the questions.

- **Manual review.** We assess the questions for accuracy, ensuring they are factually correct and relevant to our purpose. Manual reviews can also provide insights into whether the questions are likely to effectively elicit the intended information from answers, thereby contributing to the reliability and validity of the benchmark.

## 6 EXPERIMENT

### 6.1 EXPERIMENT SETUP

**Datasets.** We utilize the training set of MEGraphQA for fine-tuning, and its test set is employed for evaluation.

Table 3: Experimental results for MEGraphQA.

| Models | Accuracy | | | |
| --- | --- | --- | --- | --- |
| | Comparison | Statistics | Relationship | Overall |
| GPT-3.5-turbo | 0.105 | 0.198 | 0.476 | 0.239 |
| GPT-3.5-turbo + RAG | 0.605 | 0.260 | 0.476 | 0.425 |
| GPT-4 | 0.199 | 0.289 | 0.507 | 0.316 |
| GPT-4 + RAG | 0.763 | 0.410 | 0.687 | 0.593 |
| Llama-3-Instruct | 0.046 | 0.118 | 0.256 | 0.130 |
| Llama-3-Instruct + RAG | 0.447 | 0.181 | 0.410 | 0.325 |
| FT Llama-3-Instruct | 0.046 | 0.253 | 0.259 | 0.189 |
| FT Llama-3-Instruct + RAG | 0.687 | 0.448 | 0.573 | 0.556 |
| **SQA (Ours)** | **0.934** | **0.908** | **0.803** | **0.889** |

**Baselines.** For open-source LLMs, we conduct experiments using the representative Meta-Llama-3-8B-Instruct (Meta Llama3, 2024) and further apply QLoRA (Dettmers et al., 2023) to fine-tune it with the training set. For proprietary LLMs, we select the widely acknowledged GPT models, including GPT-3.5-turbo (Ouyang et al., 2022) and GPT-4 (Achiam et al., 2023), which have been extensively adopted in various applications. Additionally, we incorporate RAG across all vanilla baseline models for comparative analysis and evaluating the model's capacity to integrate and leverage external data sources.

**Evaluation Metrics.** We adopt Accuracy ($Acc$) as the primary metric to assess the performance of LLMs on MEGraphQA tasks. For the subcategories of Variance Analysis, Correlation Analysis, and Distribution Compliance within the Statistics tasks shown in Table 1, we focus solely on prompting LLMs to identify relevant columns and applicable methods, evaluating the accuracy of their selections instead of the computational results, as LLMs' abilities in precise calculations are not the central focus of this study.

## 6.2 RESULTS AND ANALYSIS

Different models display significant performance differences on MEGraphQA, showing sufficient discriminative ability. Table 3 presents experimental results on three categories of queries alongside overall accuracy on MEGraphQA, and Figure 3 shows accuracy on eight further-divided sub-types.

**Performance of SQA and SOTAs.** Compared to other existing methods, our SQA approach significantly enhances overall accuracy, reaching 88.9% and raising the SOTA by **29.6%**. In relation to the current leading approach (GPT-4 + RAG), our approach outperforms accuracy in the relational and comparative query types by 11.6% and 17.1%, respectively, while achieving a remarkable improvement of 46% for statistical query types.

**Baseline Performance.** For baselines, the introduction of RAG leads to impressive improvement in overall performance, especially on comparison tasks. In contrast, fine-tuning LLaMA-3-Instruct fails to yield a prominent improvement; it still requires the incorporation of RAG to achieve a fundamental performance boost. Additionally, the capabilities of the LLM itself serve as a crucial factor. On MEGraphQA, current open-source models are unable to compete with cutting-edge proprietary models like GPT-4. Even after fine-tuning and introducing RAG, LLaMA-3-Instruct can only achieve an overall accuracy comparable to that of vanilla GPT-4 (31.6%), whereas GPT-4 combined with RAG reaches the highest baseline overall accuracy of 59.3%.

**Fine-grained Performance on Sub-tasks.** Figure 3 illustrates fine-grained results of each model across eight subcategories of tasks. We observe that vanilla LLMs perform relatively well in the correlation analysis and descriptive relationship sub-tasks. Based on this, RAG demonstrates the most significant improvements in the two sub-tasks of the comparison task: intercomparison and superlative. As mentioned above, although fine-tuning does not result in a noticeable improvement in overall accuracy, it significantly strengthens the capabilities of LLaMA-3-Instruct specifically in

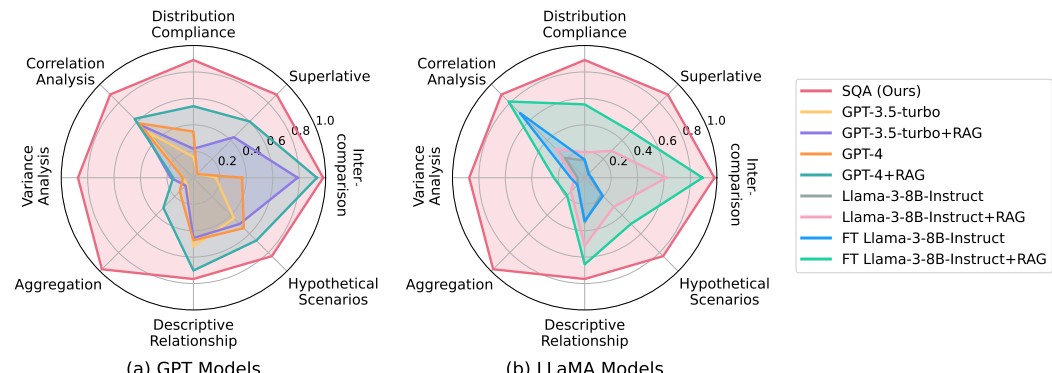

Figure 3: Experimental results for eight sub-types queries of each model.

the correlation analysis sub-task. However, neither fine-tuning nor RAG fundamentally alters the challenges faced by LLMs in the variance analysis and aggregation sub-tasks, as even GPT-4 with RAG still exhibits low accuracy in both sub-types. By comparison, our proposed SQA achieves comprehensive superiority across all sub-tasks as shown in Figure 3, attaining exceptional accuracies of 87.3% and 97.9% in variance analysis and aggregation, respectively.

**Errors Analysis for SQA.** We sample and analyze the output of SQA system, and the experimental outcomes are in line with what was anticipated, which SQA achieves remarkable performance and sets new SOTA. Just as we considered when designing our system model selection in Section 4.4, it faces two challenges which are listed below. Also, we find a new challenge with synonyms, revealing the potential for further improvement.

- **Relation semantic parsing.** The semantic parsing model of SQA effectively recognizes entities but has limited capability in relationship identification. This limitation can result in difficulties during graph retrieval, where no entities are retrieved, negatively impacting the performance of approaches with RAG including our SQA. For instance, in the query "How many US Presidents have served more than one term in office?" the model incorrectly identifies the relationship as "instance of" rather than the correct "position held", leading to erroneous results.

- **Insufficient information extraction.** We also identified errors in SQA's information extraction during the table-filling phase. An analysis of over 2,000 table-filling instances reveals that these errors primarily occur as omissions (albeit with a low probability of approximately 0.1%). A new challenge is the appearance of multi-word synonyms within the same column, like "US" and "America", which negatively affects the accuracy of SQL execution such as "SELECT".

## 6.3 FUTURE WORK

The challenges outlined in error analysis of Section 6.2 highlight the necessity for improved semantic understanding of queries and the utilization of Large Language Models (LLMs) to enhance the accuracy of data extraction. To address multi-word synonyms, one potential solution involves the incorporation of semantic operators (Patel et al., 2024), which would empower users to execute advanced operations and queries on tabular data. The integration of these operators represents a substantial progression in user interaction with and querying of tabular datasets. Specifically, these operators will be designed to grasp the intent and context of user queries, surpassing basic keyword matching to capture the subtleties of natural language. By implementing semantic operators, users will be able to conduct sophisticated operations, such as filtering by specific criteria, aggregating data according to designated attributes, and performing complex comparative analyses, all through user-friendly and intuitive queries. Addressing these challenges will be the primary focus of our future efforts to refine our methodology and enhance the system's ability to effectively process multi-entity queries.

## 7 CONCLUSION

In conclusion, our research presents a novel approach, Structured Question Answering (SQA), to address the complexities involved in multi-entity question answering (QA) from Wikipedia. Existing methods, particularly those employing retrieval-augmented generation (RAG) alongside large language models (LLMs), often fall short in effectively aggregating and reasoning over information scattered across multiple Wikipedia pages. By leveraging the inherent structure of Wikipedia's wiki-graph for multi-entity retrieval and introducing a method to organize extracted entities into a relational table format, the SQA approach significantly enhances the performance for answering multi-entity questions. The exhaustive experiments conducted as part of this study underscore the superior performance of SQA in overcoming the limitations of traditional LLM-based solutions. This research not only presents a more effective methodology for multi-entity QA but also sets the stage for future explorations into improving the accuracy and efficiency of information mining from large, unstructured knowledge bases.

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

# A  APPENDIX

## A.1  PROMPT

### A.1.1  PROMPT FOR SCHEMA

Create a table schema that comprehensively captures information about {}. Ensure the schema is detailed and structured, avoiding over-simplification, missing elements, and redundancy. This schema should be structured so each row represents a unique instance, with each column capturing a distinct aspect of property details. Ensure there is no overlap in content between columns to avoid repetition.

### A.1.2  PROMPT FOR OUTPUT CONTROL

...review your output to ensure it meets all the above criteria. Your goal is to produce a clear, accurate, and well-structured output. Just output the {}, no other word or symbol.

## A.2  TABLES

Table 4 shows examples of topics and their entities' properties.

Table 5 shows examples of question templates to synthesize queries.

Table 4: Example Topics and Their Entities Properties.

| Topics | Entities Properties | #-Entities |
|---|---|---|
| ACM fellow | nationality, field of study, affiliation | 1115 |
| Cities of the World | population, geographic coordinates, altitude, GDP | 7040 |
| Presidents of the US | term lengths, political parties, vice-presidents, birth states, previous occupations | 55 |
| Chemical Elements | atomic number, atomic mass, boiling point, melting point, electron configuration | 166 |
| Summer Olympic Games | host cities, number of participating countries, total number of events, medal tally, records broken | 35 |
| Nobel Prize in Chemistry | categories, year of award, country of origin, field of contribution. | 194 |

Table 5: Template example for queries generated by the LLM (GPT-4).

| Types | Sub-types | Template Examples |
|---|---|---|
| Comparison | Intercomparison | Which has high [property], [entity A] or [entity B]? |
| | Superlative | Which [entity] has the highest/lowest [property]? |
| Statistics | Aggregation | How many [entities] have [specific property value]? |
| | Distribution Compliance | Does [property] follow a normal distribution? |
| | Correlation Analysis | Is there a linear relationship between [property A] and [property B]? |
| | Variance Analysis | Are the variances in [property A] and [property B] significantly different? |
| Relationship | Descriptive Relationship | How is [entity A] related to [entity B]? |
| | Hypothetical Scenarios | What would be the impact if [entity A] collaborates with [entity B]? |

