# OpenReview forum: "Harnessing the Wikipedia Graph for Effective Multi-Entity Question Answering"
_ICLR.cc/2025/Conference — ICLR 2025 Conference Withdrawn Submission_

### Official Review · Reviewer_tgBr · 2024-10-31

**Soundness:** 1
**Presentation:** 1
**Contribution:** 1
**Rating:** 3
**Confidence:** 3

**Summary:**

The paper introduces a dataset of about 4780 question answers based on Wikipedia. The questions involve multiple entities and can be classified into three broad categories: Comparison, Statistical, and Relationship questions. The paper also proposes an approach called SQA to answer such questions. The key idea is to create an intermediate table that summarizes the information needed to answer the question.

**Strengths:**

The paper proposes a method for solving the difficult task of answering non-trivial questions involving multiple entities. However, due to the shortcomings mentioned in the weakness section, it is difficult to assess the paper's significance based on the current draft.

**Weaknesses:**

**Clarity**: The paper lacks clarity and purpose. Therefore, in its current form, it may not be useful to the community.
* There are several places where sufficient information is not provided to get a point through.  For example, in section 2.3, the authors cite a couple of papers stating that those papers propose GraphQA benchmarks. The authors then simply go on to say that those benchmarks are insufficient without providing any specific reason. In section 4.1, the authors mention that they use named entity recognition and relation extraction models but fail to provide any details of the model or a reference to a pre-existing model they use for the purpose. Another example is section 5.2, which tries to describe the process of generating QA pairs to create the dataset. From the two paragraphs in the section (lines 306 to 320), I was unable to understand how they created the QA pairs.

**Quality**: The experiments are insufficient to establish the usefulness of the dataset as well as the real efficacy of the proposed model.

**Questions:**

1. Around line 403, you mention that for some subcategories of questions under the Statistics, you do not evaluate the accuracy of the model on the numerical answers but instead check whether the SQA model can identify the relevant columns in the table that it creates. Then, for these categories, how do we evaluate the baselines? As I understand, the baselines do not create the intermediate table before answering the queries.

2. Could you please provide the details of the name entity recognition and relation extraction models you use?

3. Since the paper is highly empirical, could you please include the reproducibility statement as described in the [ICLR author instructions](https://iclr.cc/Conferences/2025/AuthorGuide).

---

### Official Review · Reviewer_sfh1 · 2024-11-02

**Soundness:** 2
**Presentation:** 1
**Contribution:** 2
**Rating:** 3
**Confidence:** 4

**Summary:**

The paper addresses multi-entity question answering (QA) using Wikipedia as a knowledge source. It argues that modern methods, particularly those involving retrieval-augmented generation (RAG) with large language models (LLMs), struggle to synthesize information from multiple Wikipedia pages effectively. Instead, it suggests an approach that constructs a table as a summary to answer multi-entity questions. The paper proposes a Structured Question Answering (SQA) approach that works in three main steps: (1) graph search to identify entities from the question, (2) table generation to extract relevant information from graphs, and (3) execution of SQL to get the answer. In order to test the proposed approach, the paper introduces MEGraphQA, a new benchmark for evaluating multi-entity QA on Wikipedia. In experiments, the SQA approach significantly outperforms existing methods.

**Strengths:**

1. The paper introduces using a table as an intermediate object for context construction.
2. The paper introduces another dataset for multi-hop reasoning question answering.

**Weaknesses:**

1. While the overall idea of the SQA approach is straightforward and original, its description is rather vague.
    1. In Section 4.1, the paper describes the first step to parse the question into entities and relations. The paper mentions the use of GPT-4, NER, and RE. However, no details are given on how these are achieved.
    2. In Section 4.2, the paper describes the schema generation prompts in the appendix but does not state how Mistral-7B populates the table.
    3. In Section 4.3, there is no detail of how a question is converted to SQL.
2. The paper should discuss many existing multi-hop reasoning datasets to motivate another benchmark. For example, this article reviews seven multi-hop QA datasets ([ref](https://arxiv.org/pdf/2204.09140#page=27.11)). In addition, the paper mentions a quality control step in Section 5.3 but does not provide any detail or evidence that it is done thoroughly.
3. The comparison in Table 3 shows that SQA is better than RAG. However, some important, missing baselines, such as [StructQA](https://arxiv.org/abs/2311.03734) or [HOMLES](https://aclanthology.org/2024.acl-long.717/), might be missing.
4. The paper could benefit from writing improvement. Several paragraphs do not convey important reasoning or evidence but rather an opinionated advantage of the approach, such as Section 4.4, Lines 103 - 107, and Lines 358 - 362. Some minor comments about the notations include notations not being used, unclear what $P$ and $t$ are, duplicate duplicates of $V$, and lack of definition of $P_v$.

**Questions:**

-

---

### Official Review · Reviewer_LFcr · 2024-11-02

**Soundness:** 3
**Presentation:** 3
**Contribution:** 2
**Rating:** 3
**Confidence:** 3

**Summary:**

Wikipedia is widely used for QA, but traditional methods struggle with multi-page information. This paper proposes a Structured QA (SQA) approach, organizing entities into relational tables and using table-based QA, improving accuracy by 29.6% over current methods.

**Strengths:**

1, The structure of the presentation is clear, covering the whole process from classified query and graph search to table generation and execution.
2.  This paper  constructed a specialized benchmark for Multi-Entity QA. However, no access link is provided, so it is not possible to judge its quality.
3.  The SQA approach,  using a relational table to structure multi-entity data, aims to enhancethe retrieval accuracy and scalability of question answering over Wikipedia, which is reasonable.

**Weaknesses:**

1. The core idead in  this paper, i.e., using a relational table to structure multi-entity data,  has been studied in work "Semantic Table Retrieval Using Keyword and Table Queries", "Web Table Extraction, Retrieval, and Augmentation: A Survey", "Novel Entity Discovery from Web Tables" and "Web Table Extraction, Retrieval and Augmentation".
2. The question and answer is based on the contents of the relational table. How to ensure the accuracy of the entity relations extracted by GPT-4? it has not been explained whether there is a more efficient way to build structured entity relations, and whether the additional time and space complexity introduced by the construction table.
3. The performance achieved is heavily dependent on GPT-4, and thus the availability is relatively limited. Therefore, it is recommended that the author utilize other open-source large models as tools to replace GPT-4 for comparison with existing methods, in order to verify the validity of the core idea.

**Questions:**

1. The core idead in  this paper, i.e., using a relational table to structure multi-entity data,  has been studied in work "Semantic Table Retrieval Using Keyword and Table Queries", "Web Table Extraction, Retrieval, and Augmentation: A Survey", "Novel Entity Discovery from Web Tables" and "Web Table Extraction, Retrieval and Augmentation".
2. The question and answer is based on the contents of the relational table. How to ensure the accuracy of the entity relations extracted by GPT-4? it has not been explained whether there is a more efficient way to build structured entity relations, and whether the additional time and space complexity introduced by the construction table.
3. The performance achieved is heavily dependent on GPT-4, and thus the availability is relatively limited. Therefore, it is recommended that the author utilize other open-source large models as tools to replace GPT-4 for comparison with existing methods, in order to verify the validity of the core idea.

---

### Official Review · Reviewer_o1jz · 2024-11-03

**Soundness:** 2
**Presentation:** 3
**Contribution:** 2
**Rating:** 5
**Confidence:** 5

**Summary:**

This paper introduces MEgraph QA, a benchmark targeting multi-entity question answering, and proposes the Structured QA framework to address this challenge. The framework’s core innovation is converting long texts into tables, effectively compressing input length and filtering irrelevant information. Structured QA demonstrates optimal performance on the MEgraph QA benchmark.

**Strengths:**

1.	The benchmark construction process is clear. The author describes the benchmark construction process in detail, including question templates. The dataset is also analyzed in detail, including the types of questions, entity numbers, etc.
2.	The motivation is straightforward. Excessive retrieved text often introduces irrelevant information, yet the authors effectively improve answer accuracy by extracting key information into a table to filter the noise.

**Weaknesses:**

1.	There is a weak correlation between the problem and the solution. While the authors focus on multi-entity question answering, the method primarily addresses the simplification of long retrieval documents. In this sense, the proposed method does not need to focus on entities but can be applied in any scenario where the retrieval text is too long, but the authors do not experimentally validate it.
2.	The significance of the benchmark remains unverified. In the main experiments, the authors evaluate only the base model, RAG, and Structure QA, without exploring alternative RAG methods [1] or other graph-based approaches[2,3]. We do not know if the benchmark is necessary for further research.
3.	The effectiveness of the proposed method is not convincing. As noted in W1 and W2, SQA in the main experiment is only compared with basic methods, Specifically, the experiment employs the highly capable GPT-4 as the main model of SQA (refer to lines 212 and 252), resulting in a potentially imbalanced comparison with less advanced models such as GPT-3.5 and Llama. We hope that the authors can add more baseline[1,2,3].
4.	The authors should consider conducting ablation experiments to verify the necessity of SQL in their approach. A comparative analysis between the SQL query-based table method and simpler techniques, such as keyword extraction and summarization, would clarify the relative advantages of SQL in this context.

[1] Wang H, Li R, Jiang H, et al. Blendfilter: Advancing retrieval-augmented large language models via query generation blending and knowledge filtering[J]. arXiv preprint arXiv:2402.11129, 2024.
[2] Wu Y, Huang Y, Hu N, et al. CoTKR: Chain-of-Thought Enhanced Knowledge Rewriting for Complex Knowledge Graph Question Answering[J]. arXiv preprint arXiv:2409.19753, 2024.
[3] Sun J, Xu C, Tang L, et al. Think-on-Graph: Deep and Responsible Reasoning of Large Language Model on Knowledge Graph[C]//The Twelfth International Conference on Learning Representations.

**Questions:**

1.	Can you describe in detail how to use the entities and relations in the text to query the graph in Figure 1-a3? Is it query by string matching?
2.	For the executor in Section 4.3, how do LLM and SQL interact with each other? Does LLM write the SQL query and then execute it?
3.	For Section 5.3, why is quality control necessary? In practice, the semantics of the questions are usually unclear.

---

### Official Review · Reviewer_7eqN · 2024-11-04

**Soundness:** 2
**Presentation:** 2
**Contribution:** 2
**Rating:** 3
**Confidence:** 4

**Summary:**

The paper addresses the challenge of multi-entity question answering (QA) on Wikipedia, where queries often involve complex relationships across multiple entities, like “How many Turing Award winners are Canadian?” Traditional approaches, such as using retrieval-augmented generation (RAG), struggle with aggregating insights from multiple pages effectively. To overcome this, the authors propose a Structured Question Answering (SQA) approach that organizes information into relational tables tailored to each query. This structure enables advanced table-based methods, like TableQA or NL2SQL, for more accurate responses.

**Strengths:**

1. The development of MEGraphQA as a benchmark specifically for multi-entity QA over Wikipedia is a notable contribution.
2. The paper explores a practical area of application by focusing on Wikipedia, a widely used knowledge base, for QA. This emphasis on a real-world source may make the research outcomes directly relevant to practical implementations of QA systems.

**Weaknesses:**

1. Many KBQA systems already handle multi-entity questions effectively, utilizing pre-existing data structures within the knowledge base to support relational queries.  The paper does not convincingly establish why a dedicated approach, such as SQA, is necessary.
2. The idea of converting entity data into structured tables is well-established in database query systems, and approaches like NL2SQL have long enabled structured QA without the need for a specialized framework.  Thus, SQA lacks originality and does not present a significant innovation in the context of established KBQA solutions.
3. The description of the SQA approach lacks depth in explaining how the schema is derived or validated, how the table generation process addresses entity disambiguation, and the specific steps of entity extraction. A more detailed breakdown of each process would help clarify the novelty and complexity of SQA.
4. The paper primarily discusses simple schema extraction for relatively straightforward queries. However, in real-world applications, queries can vary widely in complexity, requiring more sophisticated relational or semantic inference.   The paper could be strengthened by addressing how SQA would handle more complex or ambiguous queries.

**Questions:**

1. How does the proposed system ensure the correctness of the inferred schema, especially in complex or ambiguous queries? What mechanisms are in place to handle errors in schema derivation or entity disambiguation?
2. Does SQA encounter any specific failure modes or limitations in terms of entity coverage, complex relationship resolution, or accuracy? How does it handle such limitations, and are there planned mitigations?

---

### Note · Authors · 2024-11-21

I have read and agree with the venue's withdrawal policy on behalf of myself and my co-authors.